# Multi-Source Partial Discharge Fault Location with Comprehensive Arrival Time Difference Extraction Method and Multi-Data Dynamic Weighting Algorithm

**DOI:** 10.3390/e25040572

**Published:** 2023-03-27

**Authors:** Disheng Wang, Lin Du, Tao Wang, Xiuna Zhao

**Affiliations:** 1School of Electrical Engineering and Electronic Information, Xihua University, Chengdu 610097, China; 2School of Electrical Engineering, Chongqing University, Chongqing 400044, China

**Keywords:** ultra-high frequency, multi-source partial discharge location, energy accumulation, secondary correlation, dynamic weighting

## Abstract

The location of the partial discharge source is an important part of fault diagnosis inside power equipment. As a key step of the ultra-high frequency location method, the extraction of the time difference of arrival can generate large errors due to interference. To achieve accurate time difference extraction and further multi-source partial discharge location, a location method with comprehensive time difference extraction and a multi-data dynamic weighting algorithm is proposed. For time difference extraction, the optimized energy accumulation curve method applies wavelet transform and mode maximization calculations such that it overcomes the effect of interference signals before the wave peak. The secondary correlation method improves the interference capability by performing two rounds of correlation calculations. Both extraction methods are combined to reduce the error in time difference extraction. Then, the dynamic weighting algorithm effectively utilizes multiple data and improves the location accuracy. Experimental results on multi-source partial discharge locations performed in a transformer tank validate the accuracy of the proposed method.

## 1. Introduction

Partial discharge (PD) is a common fault in the internal insulation of power equipment. The PD source location is an important part of PD diagnosis. Conventional PD source location methods include acoustic, optical, electrical and ultra-high frequency (UHF) methods [1,2,3,4,5]. Because of its high sensitivity and interference suppression, the UHF method has been widely studied and applied. The accuracy of the time difference of arrival (TDOA) extraction is an important factor in the location effect of the UHF method [6]. TDOA extraction methods for UHF signals mainly include the first peak method, the wavelet transform method, the cross-correlation method and the energy accumulation method [7,8,9].

The first peak method uses the starting point or some threshold value of the first pulse peak as a reference point to calculate the TDOA [10,11]. When the first peak protrudes, this method is convenient with high accuracy [7]. Unfortunately, the interference signal before the first peak can affect the determination of the reference point. The wavelet transform method is performed by applying a wavelet transform to the UHF signal and determining the mode maximum point as the reference point [12,13]. However, interference-generated pulse peaks can easily lead to a shift of the mode maximum point, and a large TDOA error may still exist. The cross-correlation method establishes the cross-correlation function of two UHF partial discharge signals [14,15]. Then, when the cross-correlation function reaches its peak value, the time-independent variable corresponding to the peak point is equal to the TDOA. If there is an interference signal similar to the discharge pulse in the UHF signal, then the cross-correlation function may have multiple peaks, resulting in difficulty in determining the TDOA. The energy accumulation method reduces the effect of interference signals by converting UHF partial discharge signals into energy accumulation curves. Existing studies take the starting point of the slope of the energy accumulation curve as the reference point to calculate TDOA [8,16]. Although the energy accumulation method is the most robust to noise from single PD sources [8], it is still susceptible to large interference signals before the first peak.

In addition, the multi-source partial discharge (MSPD) location is more difficult than that of a single PD source. When there are multiple PD sources in space-limited power equipment, the UHF signals generated by different PD sources interfere with each other. At the same time, signals propagating in other directions can also begin to interfere after reflecting on structures such as metal tank walls. To distinguish UHF signals from different PD sources, an additional acoustic measurement is adopted [10,17], which increases measurement complexity. In [18], the MSPD location has been realized based on the simulation model and the TDOA database. However, it is not convenient to use because it only applies to specific equipment.

To address the issues described, this paper proposes a PD location method for MSPD inside power equipment by integrating two TDOA extraction methods and using multi-data dynamic weighting. The proposed PD fault location method has the following salient characteristics:(1)The optimized energy accumulation method and the secondary correlation method are proposed and combined to reduce the error in TDOA extraction. In addition, dynamic weighting algorithms are used to effectively utilize multiple groups of TDOA data in PD source calculations and improve the location accuracy. Compared with the other methods, the localization method has better interference and accuracy.(2)Compared to the conventional method, the optimized energy accumulation curve method obtains the inflection point of the energy curve as a reference point through wavelet transform and mode maximization calculations, which overcomes the effect of the interference signal before the wave peak.(3)In the secondary correlation method, the effect of interfering signals is mitigated by the two rounds of correlation calculations. The interference capability of TDOA extraction is further improved compared to the cross-correlation method.

The remainder of this paper is organized as follows. Two TDOA extraction methods are described in Section 2. Practical applications of these two methods to TDOA extraction of the MSPD are described, and their comprehensive application strategy is provided in Section 3. Following, the dynamic weighting algorithm for multiple TDOA data is proposed and the experimental results of the MSPD location are analyzed in Section 4. Finally, conclusions are drawn in Section 5.

## 2. TDOA Extraction Methods

In practice, the TDOA is the time difference between different detection sensors receiving the same UHF partial discharge signal. The PD source position can be calculated using three TDOAs. In the calculation, the UHF signals are considered to be transmitted along a straight line. Define *P* (*x*, *y*, *z*) as the PD source coordinates and *S_i_* (*x_i_*, *y_i_*, *z_i_*) as the *i*-th sensor coordinates. Taking *S*_1_ as the reference, the propagation distance difference Δ*r_i_*_1_ is:(1)$$\Delta r_{i1}=\sqrt{{\left(x-x_{i}\right)}^{2}+{\left(y-y_{i}\right)}^{2}+{\left(z-z_{i}\right)}^{2}}-\sqrt{{\left(x-x_{1}\right)}^{2}+{\left(y-y_{1}\right)}^{2}+{\left(z-z_{1}\right)}^{2}}=c\Delta t_{i1}$$
where *c* denotes the velocity of the electromagnetic wave; Δ*t_i_*_1_ denotes the TDOA between the first and the *i*-th sensors. The coordinates of *S_i_* (*x_i_*, *y_i_*, *z_i_*) and *c* are known. When more than three Δ*t_i_*_1_ are achieved, the coordinate of *P* (*x*, *y*, *z*) can be obtained by solving the equation groups.

To suppress the periodic interference or white noises, the wavelet denoising and filtering are performed on the collected UHF signals before the TDOA extraction [19,20].

### 2.1. Optimized Energy Accumulation Method

The energy accumulation curve *E* is calculated based on the UHF partial discharge signal:(2)$$E=\sum_{k=0}^{n}u_{k}{}^{2}$$
where *u_k_* denotes the voltage value of the *k*-th point in the PD signal segment, and *n* denotes the sampling point number.

Two channels of the UHF signals and corresponding energy accumulation curves are shown in Figure 1. The PD pulse period in Figure 1a corresponds to the rising segment of the energy accumulation curve in Figure 1b.

The traditional energy accumulation method takes the starting point of the curve slope as the reference point to calculate the TDOA [12]. This is easily affected by interference signals before the first wave peak.

In this paper, the wavelet modulus maximum calculation is employed to detect singularities so that the inflection point of the slope of the curve can be determined. The inflection point of the curve slope is then used as a reference point to extract the TDOA.

Wavelet transform is applied firstly to smooth the energy accumulation curve *f*(*x*) to reduce the impact of interference signals and noises. The cubic B-spline function *θ*(*x*) is applied in this paper, which can be regarded as the response function of the low-pass filter and can be approximated as:(3)$$\hat{\theta }\left(x\right)={\left(\frac{sin\left(\frac{x}{4}\right)}{\frac{x}{4}}\right)}^{4}$$

Define $\varphi^{\prime }(x)=d\theta (x)/dx$, and then perform the convolution calculation with *f*(*x*) and $\varphi^{\prime }(x)$. The calculated wavelet transform function is $W^{\prime }f_{s}\left(x\right)$, which can be expressed as:(4)$$W^{\prime }f_{s}\left(x\right)=f\left(x\right)⊗\varphi^{\prime }\left(x\right)=f\left(x\right)⊗\left(\frac{d\theta \left(x\right)}{dx}\right)=\frac{d}{dx}\left(f⊗\theta \right)\left(x\right),\;x\in s$$
where ⊗ denotes convolution calculation, and *s* denotes the transform scale. $W^{\prime }f_{s}\left(x\right)$ is proportional to the first derivative of smoothed *f*(*x*).

Then the inflection point of the slope of the energy accumulation curve can be determined by calculating the partial modulus maximum of the absolute value of $W^{\prime }f_{s}\left(x\right)$. Figure 2 shows the wavelet modulus maximum lines corresponding to the energy accumulation curves in Figure 1. It can be seen that the modulus maximum line for the first wave peak of the PD pulse is significantly higher than the lines for the subsequent wave peaks and noises. Therefore, the time point corresponding to the modulus maximum line with the highest amplitude can be used as a reference point to calculate the TDOA. The optimized energy accumulation method overcomes the effect of the interference signal before the wave peak and achieves high accuracy when the PD pulse is prominent.

### 2.2. Secondary Correlation Method

The UHF partial discharge signals of two sensors from the same PD source are recorded as *s*_1_(*t*) and *s*_2_(*t*), respectively, which are expressed as:(5)$$\left\{ \begin{array}{l} s_{1}\left(t\right)=a_{1}\cdot s\left(t-t_{1}\right)+n_{1}\left(t\right)\\ s_{2}\left(t\right)=a_{2}\cdot s\left(t-t_{2}\right)+n_{2}\left(t\right) \end{array}\right.$$
where *s*(*t*) denotes signal of the PD source; *a*_1_ and *a*_2_ denote attenuation constants, respectively; *t*_1_ and *t*_2_ denote time differences, Δ*t* = *t*_2_ − *t*_1_ denotes the TDOA of the two signals; *n*_1_(*t*) and *n*_2_(*t*) are signal distortions and noises, respectively.

Employ the self-correlation and the cross-correlation calculation to signals *s*_1_(*t*) and *s*_2_(*t*). The self-correlation function $R_{s_{1}s_{1}}(\tau )$ and cross-correlation function $R_{s_{1}s_{2}}(\tau )$ are:(6)$$R_{s_{1}s_{1}}\left(\tau \right)=\int_{-\infty }^{+\infty }s_{1}\left(t\right)\cdot s_{1}\left(t+\tau \right)dt=E\left[s_{1}\left(t\right)s_{1}\left(t+\tau \right)\right]$$
(7)$$R_{s_{1}s_{2}}\left(\tau \right)=\int_{-\infty }^{+\infty }s_{1}\left(t\right)\cdot s_{2}\left(t+\tau \right)dt=E\left[s_{1}\left(t\right)s_{2}\left(t+\tau \right)\right]$$

The traditional cross-correlation method extracts the TDOA by calculating the peak value of the cross-correlation function. According to Equations (5) and (7), the cross-correlation function $R_{s_{1}s_{2}}(\tau )$ is calculated as:(8)$$\begin{array}{ll} R_{s_{1}s_{2}}\left(\tau \right) & =E\left[s_{1}\left(t\right)s_{2}\left(t+\tau \right)\right]\\ & =E\left\{\left[a_{1}s\left(t-t_{1}\right)+n_{1}\left(t\right)\right]\cdot \left[a_{2}s\left(t-t_{2}+\tau \right)+n_{2}\left(t+\tau \right)\right]\right\}\\ & =a_{1}a_{2}R_{ss}\left(\tau -\Delta t\right)+a_{2}R_{n_{1}s}\left(\tau -t_{2}\right)+a_{1}R_{sn_{2}}\left(\tau +t_{1}\right)+R_{n_{1}n_{2}}\left(\tau \right) \end{array}$$
where $R_{n_{1}n_{2}}(\tau )\;\approx \;0$. If *s*(*t*) is unrelated to *n*_1_(*t*) and *n*_2_(*t*), then:(9)$$R_{s_{1}s_{2}}\left(\tau \right)=a_{1}a_{2}R_{ss}\left(\tau -\Delta t\right)$$

In Equation (8), when *τ* = Δ*t*, *R_ss_* reaches its maximum, so does $R_{s_{1}s_{2}}(\tau )$. Thus, Δ*t* can be determined by:(10)$$\Delta t=\arg\left\{max\left[R_{ss}\left(\tau -\Delta t\right)\right]\right\}=\arg\left\{max\left[R_{s_{1}s_{2}}\left(\tau \right)\right]\right\}$$
where *arg*() denotes taking independent variables.

In practice, the cross-correlation function may have multiple peaks due to attenuation, oscillation of the UHF signal and the effect of interference. That is, $a_{2}R_{n_{1}s}(\tau -t_{2})+a_{1}R_{{\mathrm{sn}}_{2}}(\tau +t_{1})$ is large. In this case, the first peak of the PD pulse is not prominent, making the reference point difficult to determine.

To improve the resolution ratio of the first peak and reduce the effect of interference, a secondary correlation method is proposed.

Define the reference signal *s*_1_(*t*) and perform self-correlation calculation to *s*_1_(*t*). The self-correlation function $R_{s_{1}s_{1}}(\tau )$ is expressed as:(11)$$\begin{array}{ll} R_{s_{1}s_{1}}\left(\tau \right) & =E\left[x\left(t\right)x\left(t+\tau \right)\right]\\ & =E\left\{\left[a_{1}s\left(t-t_{1}\right)+n_{1}\left(t\right)\right]\cdot \left[a_{1}s\left(t-t_{1}+\tau \right)+n_{1}\left(t+\tau \right)\right]\right\}\\ & =a_{1}{}^{2}R_{ss}\left(\tau \right)+a_{1}\left[R_{sn_{1}}\left(\tau +t_{1}\right)+R_{n_{1}s}\left(\tau -t_{1}\right)\right]+R_{n_{1}}{}_{n_{1}}\left(\tau \right) \end{array}$$

Then employ cross-correlation calculation with reference signals *s*_1_(*t*) and *s*_2_(*t*), as in Equation (8). Since $R_{s_{1}s_{1}}(\tau )$ and $R_{s_{1}s_{2}}(\tau )$ are functions with *τ* as the independent variable, *τ* can be substituted by *t*.

Then, perform the cross-correlation calculation with $R_{s_{1}s_{1}}(t)$ and $R_{s_{1}s_{2}}(t)$ and obtain the secondary cross-correlation function $R_{R_{1}R_{2}}(\tau )$, in which the secondary correlation calculation of the first-round correlation functions (such as $R_{\mathrm{ss}}(t)$ and $R_{sn_{1}}(t)$) will appear. In $R_{s_{1}s_{1}}(t)$ and $R_{s_{1}s_{2}}(t)$, according to Equations (8) and (11), $R_{n_{1}n_{1}}(t)$ are unrelated to other correlation functions. If $R_{\mathrm{ss}}(t)$, $R_{sn_{1}}(t)$ and $R_{sn_{2}}(t)$ are unrelated, then the cross-correlation functions (such as $E[R_{\mathrm{ss}}(t-∆t)R_{sn_{1}}(t+t_{1})]$ and $E[R_{\mathrm{ss}}(t)R_{sn_{2}}(t+t_{1})]$) are equal to zero. Thus, the secondary cross-correlation function $R_{R_{1}R_{2}}(\tau )$ can be simplified as:(12)$$\begin{array}{ll} R_{R_{1}R_{2}}\left(\tau \right) & =E\left[R_{s_{1}s_{1}}\left(t\right)R_{s_{1}s_{2}}\left(t\right)\right]\\ & =a_{1}{}^{3}a_{2}E\left[R_{ss}\left(t\right)R_{ss}\left(t-\Delta t+\tau \right)\right]+a_{1}{}^{2}a_{2}E\left[R_{ss}\left(t\right)R_{n_{1}s}\left(t-t_{2}+\tau \right)\right]+\cdots \cdots \\ & =a_{1}{}^{3}a_{2}E\left[R_{ss}\left(t\right)R_{ss}\left(t-\Delta t+\tau \right)\right]=a_{1}{}^{3}a_{2}R_{ss,ss}\left(\tau -\Delta t\right) \end{array}$$
where *R_ss_*_,*ss*_(*τ*) denotes the self-correlation function of *R_ss_*(*t*). Then, Δ*t* can be expressed as:(13)$$\Delta t=\arg\left\{max\left[R_{ss,ss}\left(\tau -\Delta t\right)\right]\right\}=\arg\left\{max\left[R_{R_{1}R_{2}}\left(\tau \right)\right]\right\}$$

In the secondary correlation method, the weight of original signal *s*(*t*) is strengthened, while the influence of noises *n*_1_(*t*) and *n*_2_(*t*) are weakened in the first round of correlation calculations. The effect of $R_{sn_{1}}(\tau )$ and $R_{sn_{2}}(\tau )$ are weakened in the second round of the correlation calculation.

### 2.3. Preliminary Comparison of TDOA Extraction Method

A preliminary comparison of the extraction effects of different extraction methods is presented. Experiments are carried out in laboratory conditions and in air, respectively. The PD sources are modeled by a needle-plate PD model. The UHF sensors adopt micro-strip antennas, whose absolute bandwidth is 340~440 MHz when the standing wave ratio (SWR) is lower than 2, and the comparative bandwidth reaches 25.6% [21]. The applied oscilloscope is the LeCroy7100, which features a simulation bandwidth of 1 GHz and a maximum sampling rate of 20 GS/s.

In the experiments, the distance between the reference sensor and the PD source is set as 1 m. Different values are used for the distance difference between the sensors. Multiple repeated experiments are performed under each set of distance differences. The TDOAs are extracted separately using the first peak method, the secondary correlation method and optimized energy accumulation method, respectively. The mean values of the TDOAs and their errors compared to theoretical values are calculated. As the results in Table 1 show, in the presence of air and without significant interference, the error of the optimized energy accumulation method and the secondary correlation method are, in general, smaller than that of the first peak method. Moreover, the optimized energy accumulation method has the highest accuracy.

## 3. MSPD TDOA Extraction

The PD locations are usually used to diagnose PD faults in power equipment such as transformers. Within a piece of power equipment, electromagnetic waves reflect off tank walls and other structures and form interference sources. Meanwhile, in the situation of multiple localized discharge sources, the signals of the discharge pulses interfere with each other. To further study the practical application of the proposed TDOA extraction method and put forward the optimal application strategy, an experimental platform is built on the basis of a real transformer tank (5.2 × 2.6 × 1.8 m). The experimental platform diagram is shown in Figure 3, and the transformer tank is shown in Figure 4a.

The needle-plate PD models are used for the PD sources to simulate PD caused by metal protrusions in a transformer, as shown in Figure 4b. The sensors use the UHF Archimedean spiral antennas [22], as shown in Figure 4c. Sensors are mounted on different surfaces of the tank. *S_i_* denotes the *i*-th sensor. Taking the corner of the tank as the origin, a virtual three-dimensional coordinate system is established. Consequently, the sensor coordinates are *S*_1_ (0.2 m, 0, 0.2 m), *S*_2_ (5.2 m, 1.8 m, 1 m), *S*_3_ (0, 1.3 m, 0.1 m) and *S*_4_ (5 m, 2.6 m, 1.8 m), respectively. The PD source coordinates are *P*_1_ (0.2 m, 0.3 m, 0.2 m) and *P*_2_ (5 m, 2.4 m, 1.8 m), respectively.

### 3.1. Application of Optimized Energy Accumulation Method

The application of the optimized energy accumulation method to the MSPD case is based on time window scanning. In this paper, the width of the main PD pulse (defined by 30% of the peak value) is about 13 ns, as shown in Figure 5, where *l*_1_, *l*_2_ and *l*_3_ denote the length, width and height of the tank, respectively. The theoretical maximum TDOA of the signals from the same PD source is $∆t_{\max}=\sqrt{{l_{1}}^{2}+{l_{2}}^{2}+{l_{3}}^{2}}/c$ ≈ 20 ns, where *c* denotes the propagation velocity of the electromagnetic wave in air and its approximate value is 30 cm/ns.

Based on the main pulse width and the theoretical maximum TDOA, the time window and scanning time step are set as 40 and 10 ns, respectively. After a discharge, the synchronization signals collected by the four sensors are shown in Figure 6.

First, for the UHF signal in a time window of 0–40 ns, the energy accumulation curves are conversed. The wavelet transform and the modulus maximization are performed. If sufficiently high modulus maximum lines do not appear simultaneously in the time window of the signals, then the time window is advanced by the scanning step of 10 ns. The above procedure is repeated for the time window of 10–50 ns. When sufficiently high modulus maximum lines appear simultaneously on all signals, record the corresponding time points.

Take a set of multi-channel synchronized signals as an example. For time window 2 shown in Figure 6, sufficiently high modulus maximum lines of all signals appear simultaneously in the 40 ns time window, as shown in Figure 7b. Since interference such as noise is weak and the main PD pulse is well-defined, each energy accumulation has only one inflection point, and the TDOAs can be computed from the corresponding time points of the modulus maximum lines.

However, for time window 1 in Figure 6, when the interferences are heavy and multiple peaks occur in a time window, several energy accumulation curves have more than one inflection point, as shown in Figure 8, leading to hard determination of the modulus maximum value lines.

Since the interference is accidental, the TDOA data can still be obtained by taking multiple sets of repeated experimental data and summarizing the calculated results. However, the TDOA extraction is less efficient. At the same time, this approach still fails to cope with the extreme case that multiple PD main pulses are exactly within the same time window.

### 3.2. Application of Secondary Correlation Method

The application of the secondary correlation method in the MSPD problem is to extract the feature signal segment from the reference PD signal and then conduct the secondary correlation with the signals of other channels.

Take the PD signals of channels 1 and 4 shown in Figure 9 as examples. The specific steps of the signal feature segment and the TDOA extraction are described as follows:

(1)Set the signal of channel 1 as the reference, which is marked as *s*_1_(*t*). First, the highest value of *s*_1_(*t*) is searched for and marked as *s*_1_(*t*_m1_). Then, search for the nearest points whose amplitude equals 30%·*s*_1_(*t*_m1_), and the corresponding time coordinates are marked as *t*_s1_, *t*_e1_. The fragment of *s*_1_(*t*) in the time interval [*t*_s1_, *t*_e1_] is defined as the first reference fragment *s*_1r1_(*t*).(2)Signal of channel 4 is the contrast signal, which is marked as *s*_4_(*t*). From *s*_4_(*t*), the contrast fragment *s*_4c1_(*t*) corresponding to the reference fragment *s*_1r1_(*t*) is selected. Considering the theoretical maximum time difference Δ*t*_max_, the time interval of *s*_4c1_(*t*) should be [*t*_s1_ − Δ*t*_max_, *t*_e1_ + Δ*t*_max_]. Then, based on Equations (11)–(13), the secondary function $R_{R_{1}R_{2}}(\tau )$ of *f*_1_(*t*) and *T*_1_(*t*) is deduced. The function curve of $R_{R_{1}R_{2}}(\tau )$ is shown in Figure 9b. The first TDOA is then extracted.(3)After completing the first TDOA extraction, set the function value of [*t*_s1_, *t*_e1_] in *s*_1_(*t*) to 0 and obtain a new reference signal *s*_1′_(*t*). Then, repeat the above steps to extract the new reference fragment *s*_1r2_(*t*) from *s*_1′_(*t*) and the contrast fragment *s*_4c2_(*t*) from *s*_4_(*t*), as shown in Figure 10. Then, the second TDOA is extracted. The TDOA extracting process above will be repeated until *s*_1_(*t*_mi_) is lower than the threshold *s_t_* = 30%·*s*_1_(*t*_m1_).

Compared to the optimized energy accumulation method, the TDOA extracted by the secondary correlation method is less affected by the interference signal. The TDOAs can also be extracted when the amplitude of the interference pulse is close to the first peak of the PD pulse.

### 3.3. Method Comparison and Application Strategy

According to the above studies, both the optimized energy accumulation method and the secondary correlation method still have weaknesses. The accuracy of the optimized energy accumulation method decreases when the amplitude of the interference pulse approaches that of the first peak. The accuracy of the secondary correlation method will still be affected by the waveform distortion of the discharge pulse.

To compare the differences between the above two extraction methods more intuitively and to make full use of them to form a complement, they are both used to extract TDOAs. Several TDOA calculation data are listed, as shown in Table 2 and Table 3.

According to Table 2 and Table 3, both the optimized energy accumulation method and the secondary correlation method are able to perform relatively accurate TDOA extraction under the conditions of internal equipment and multiple PD sources. For the optimized energy accumulation method, a part of the TDOA is not exactly solved, as shown in the blank column of Table 2.

By comparing the TDOA extraction errors of the two methods, it can be seen that when the distance between reference sensor *S*_1_ (0.2 m, 0, 0.2 m) and PD source *P*_1_ (0.2 m, 0.3 m, 0.2 m) is close, the accuracy of the optimized energy accumulation method is relatively higher. The possible reason is that, in this case, the signal from the PD source *P*_1_ has a large amplitude when it arrives at the reference sensor *S*_1_ and can be clearly distinguished from the interference signal. Therefore, the arrival time of the signal at the reference node *T*_1_ is relatively accurate, and the TDOA extraction accuracy is relatively high.

When the distance between the reference node *S*_1_ (0.2 m, 0, 0.2 m) and the PD source *P*_2_ (5 m, 2.4 m, 1.8 m) is far away, propagation of the pulse signal is relatively more disturbed. In this case, the secondary correlation method is relatively more robust against interference, and hence, the extracted TDOA is more accurate.

Based on the above analysis and the statistics of the experimental results, the strategy to apply these two TDOA extraction methods is described as follows.

(a)If the signal–noise ratio (SNR) of the PD signal is low, the secondary correlation method is used.(b)If the distance between the reference sensor and PD source is estimated to be shorter than 1 m, and SNR of the PD signal is not very low, the optimized energy accumulation method is used.(c)Otherwise, the results of the two methods should be compared.

In addition, the TDOA data with large errors can be seen in Table 2 and Table 3. Therefore, the PD sources localized by only one set of TDOAs may not be accurate.

## 4. Multi-Data Dynamic Weighting Algorithm

Multi-data statistics help to reduce the effect of accidental TDOA data and improve location accuracy [23]. In this paper, a dynamic weighting algorithm is proposed to improve the fault location accuracy for the MSPD by fully exploiting the multiple sets of the TDOAs obtained. First, the point density estimation is used to further eliminate errors due to interference. The initial points are then linearly classified into multisets. Finally, the points in each set are dynamically weighted to compute the coordinates of multiple PD sources.

### 4.1. Point Density Estimation

The PD source is defined as the primary source point. The initial source points are distributed in a 3D coordinate system based on the actual spatial size. Their densities are higher in regions close to the true PD sources. The point density estimation method is used to count the number of points in a sphere centered at a point. The density *λ*(*p*) of the initial source point *p*(*x*, *y*, *z*) is given by:(14)$$\lambda \left(p\right)=\frac{N(p,r)}{4\pi r^{3}/3}$$
where *N*(*p*, *r*) denotes the number of points involved inside the sphere, whose center is the point *p* and the radius is *r*; *4πr^3^*/3 denotes the volume of the sphere. The density threshold value *λ*_0_ will be defined according to the number of initial source points. The error initial source points (*λ*(*p*) ≤ *λ*_0_) will be removed first, such as circuited point *k* in Figure 11a.

### 4.2. Linear Classification

In addition to the error points, the number of initial source points *N*_0_ is counted. Then, the point with the highest *λ*(*p*) is searched for and defined as *p*_1_ (*x*_1_, *y*_1_, *z*_1_). *p* (*x*, *y*, *z*) denotes one of the remaining initial source points. Define the point set *S*_1_. For any point *p*, if a sphere with *p* as its center and *r* as its radius contains *p*_1_, then *p* belongs to *S*_1_. *d*(*p*_1_, *p*) denotes the distance between *p*_1_ and *p*. To facilitate linear classification, the distance constant *D* is defined, and the expression of *D* is:(15)$$\begin{array}{l} D=\frac{r^{2}-d^{2}\left(p_{1},p\right)}{r^{2}},\\ d^{2}\left(p_{1},p\right)={\left(x_{1}-x\right)}^{2}+{\left(y_{1}-y\right)}^{2}+{\left(z_{1}-z\right)}^{2} \end{array}$$

Then the relative magnitude of *d*(*p*_1_, *p*) and radius *r* can be judged by (*e^D^
*− 1). If (*e^D^
*− 1) ≥ 0, then *p* belongs to set *S*_1_.

Then, the points belonging to set *S*_1_ are removed from all initial source points. In the remaining points, search for the point with the highest *λ*(*p*) and define it as *p*_2_, and then find out the corresponding point set *S*_2_. The above linear classification procedure will be repeated until the number of unclassified points is less than the threshold *N_t_*. An example is shown in Figure 11b.

### 4.3. Dynamic Weighting

After classification, *n* point sets are obtained (*S*_1_~*S_n_*). The number of points in *S_i_* is recorded as *N_i_*. *n* high density initial source points (*p*_1_~*p_n_*) are obtained. The coordinates of *P_i_* are (*x_i_*, *y_i_*, *z_i_*). The remaining points in *S_i_* are denoted as *p_in_* (*x_in_*, *y_in_*, *z_in_*), *n* ∈ [1, *N_i_*−1]. Then the dynamic weight coefficient *α_in_* of *p_in_* is defined as:(16)$$\begin{array}{l} \alpha_{in}=\frac{1-g^{2}\left(p_{i},p_{in}\right)/G}{N_{i}-1},\;G=\sum_{n=1}^{N_{i}-1}g^{2}\left(p_{i},p_{in}\right),\;\\ g\left(p_{i},p_{in}\right)=\sqrt{r^{2}-{\left(x_{i}-x_{in}\right)}^{2}-{\left(y_{i}-y_{in}\right)}^{2}-{\left(z_{i}-z_{in}\right)}^{2}} \end{array}$$

The final PD source *P_i_* (*X_i_*, *Y_i_*, *Z_i_*) is determined by the initial source point in *S_i_*. Finally, the calculated coordinate of the PD source is:(17)$$X_{i}=\sum_{n=1}^{N_{i}-1}\alpha_{in}x_{in},\;Y_{i}=\sum_{n=1}^{N_{i}-1}\alpha_{in}y_{in},\;Z_{i}=\sum_{n=1}^{N_{i}-1}\alpha_{in}z_{in}$$

### 4.4. Application Results and Analysis

The obtained 20 sets of TDOAs (some of which are shown in Table 2 and Table 3) are used to verify the dynamic weighting algorithm. *r* is set as 5 cm, and λ_0_ is set as 2. The results of the secondary correlation TDOA extraction and dynamic weighting algorithms are taken as examples. The actual PD source (marked with red solid dots) and the calculated PD source (marked with black five-pointed stars) are shown in Figure 11c.

The detailed PD source coordinates calculated from the TDOAs obtained with both methods are shown in Table 4. The location error is within the acceptable limits considering that the propagation speed of the electromagnetic wave is 30 cm/ns.

The second set of experiments used two other PD sources *P*_3_ (220, 135, 55) and *P*_4_ (285, 155, 50), which are deeper in the tank and are close to each other. One of the measured UHF partial discharge signals is shown in Figure 12. Since the PD signal fluctuates and oscillates while the interference signal is evident, the secondary correlation TDOA extraction method is used. The location results for the second set of experiments are also presented in Table 4.

Comparing the results of the two sets of experiments with the secondary correlation method, the location error is relatively larger in the second set. In addition to the effect of PD pulse oscillations, the main causes of the errors could be the proximity of the two PD sources and the cross-influence of the UHF partial discharge signals.

Experimental results show that the proposed PD location method can maintain high accuracy when applied to the MSPD location inside power equipment with simple internal structures.

## 5. Conclusions

In this paper, a PD fault location method based on the UHF signals is proposed. To obtain accurate TDOA, the optimized energy accumulation method and the secondary correlation method are proposed, and a comprehensive application strategy is provided to fully exploit the advantages of both methods. Moreover, the dynamic weighting algorithm is used to further improve the location accuracy, and finally, the high-precision MSPD location inside the power equipment is achieved.

The main conclusions of this paper are as follows:(1)The optimized energy accumulation curve method overcomes the effect of the interference signal before the wave peak. The secondary correlation method further improves the interference capability of the TDOA extraction. Both methods have smaller errors than the first peak method for a single PD source.(2)For the MSPD location inside a piece of power equipment, the optimized energy accumulation method should be applied when the interference signal is weak and the distance between the reference sensor and the PD source is estimated to be small. The secondary correlation method should be applied when the interference signal is strong and the distance between the reference sensor and the PD source is estimated to be large.(3)The proposed dynamic weighting algorithm can fully utilize multiple TDOA data to reduce the effect of accidental data and improve location accuracy.

The UHF partial discharge signal can still be measured in the presence of internal structures, such as iron cores and windings. However, due to the obstruction of the structures, the propagation path of the electromagnetic wave is more complex, and the attenuation of the signal wave head is serious [22]. For practical application, the proposed PD location method can be further investigated by taking into account transmission velocity variations, electromagnetic wave refraction and other factors.

## Figures and Tables

**Figure 1 entropy-25-00572-f001:**
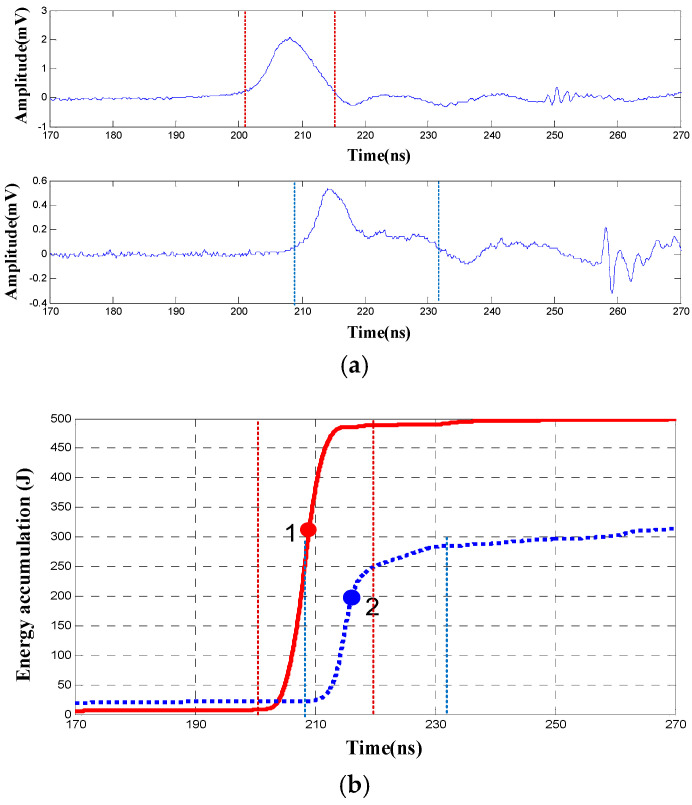
Energy accumulation curves of UHF partial discharge signals. (**a**) Group of UHF partial discharge signals in laboratory condition; (**b**) Corresponding energy accumulation curves.

**Figure 2 entropy-25-00572-f002:**
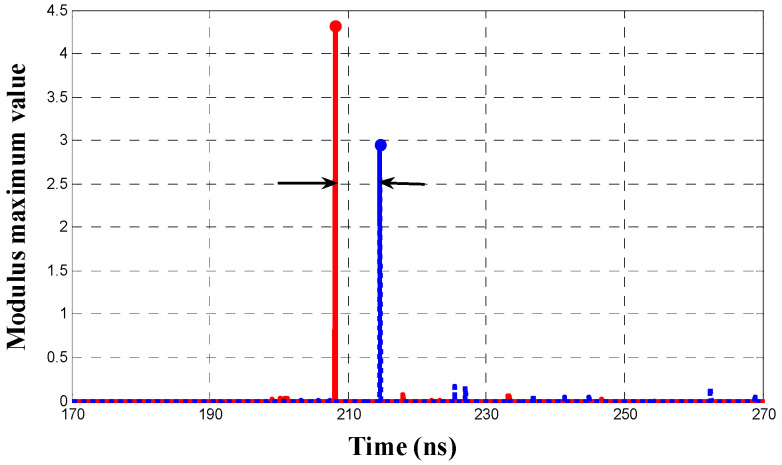
Wavelet modulus maximum lines of energy accumulation curves.

**Figure 3 entropy-25-00572-f003:**
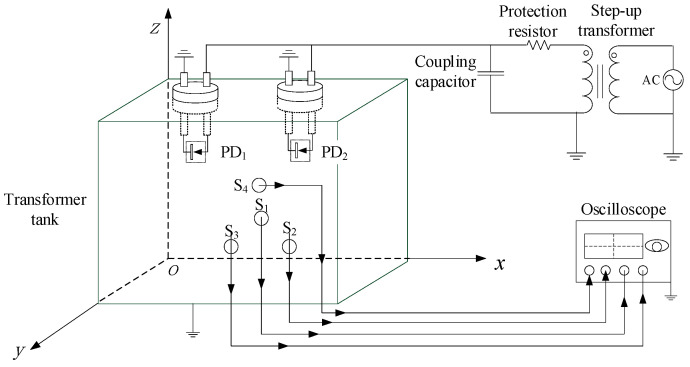
Experiment platform diagram.

**Figure 4 entropy-25-00572-f004:**
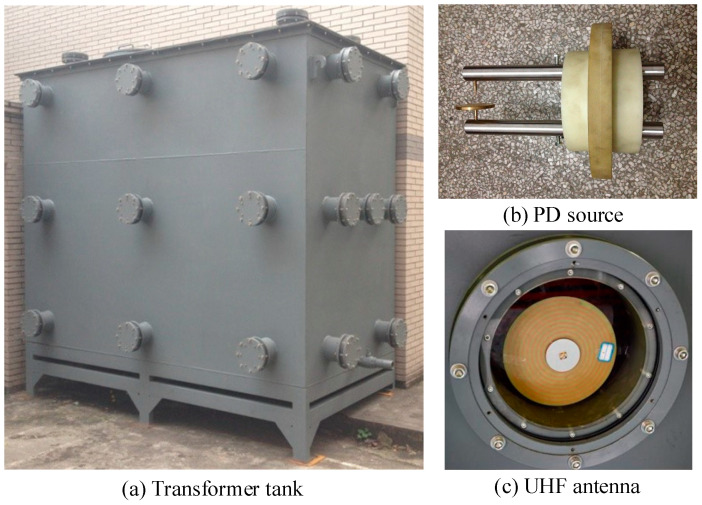
Experiment equipment.

**Figure 5 entropy-25-00572-f005:**
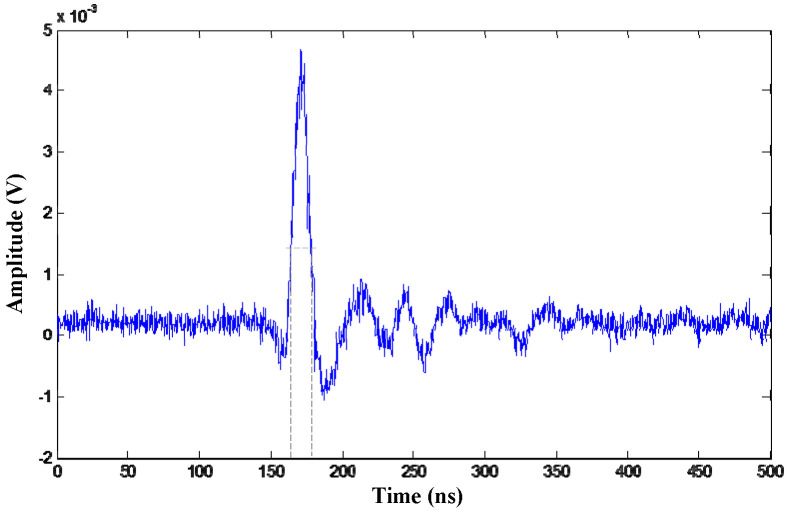
Single UHF partial discharge waveform of PD source.

**Figure 6 entropy-25-00572-f006:**
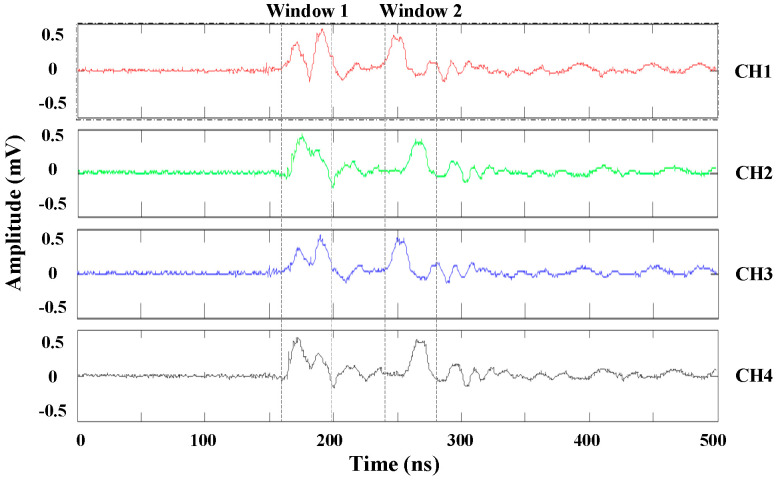
Multiple channels of UHF partial discharge pulse signals.

**Figure 7 entropy-25-00572-f007:**
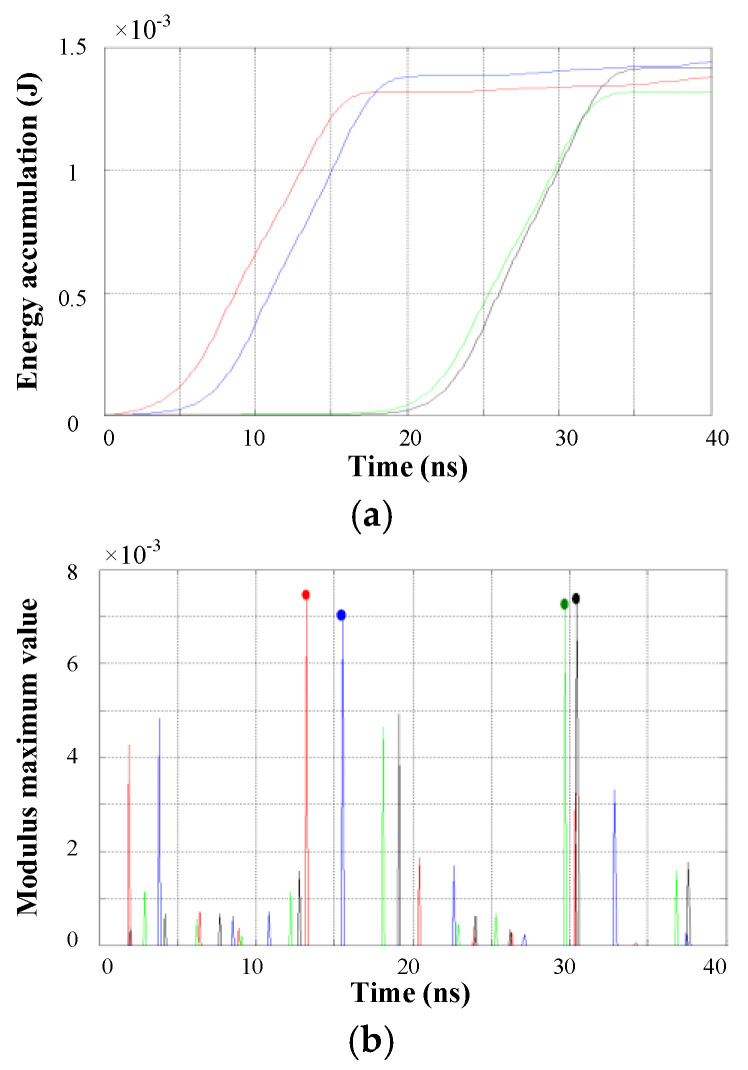
Application of energy accumulation curves method in time window 2. (**a**) Energy accumulation curves of window 2; (**b**) Wavelet modulus maximum lines.

**Figure 8 entropy-25-00572-f008:**
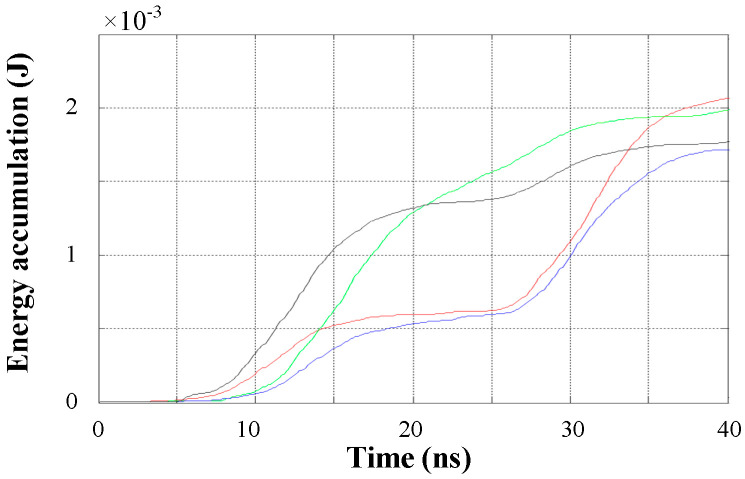
Energy accumulation curves of time window 1.

**Figure 9 entropy-25-00572-f009:**
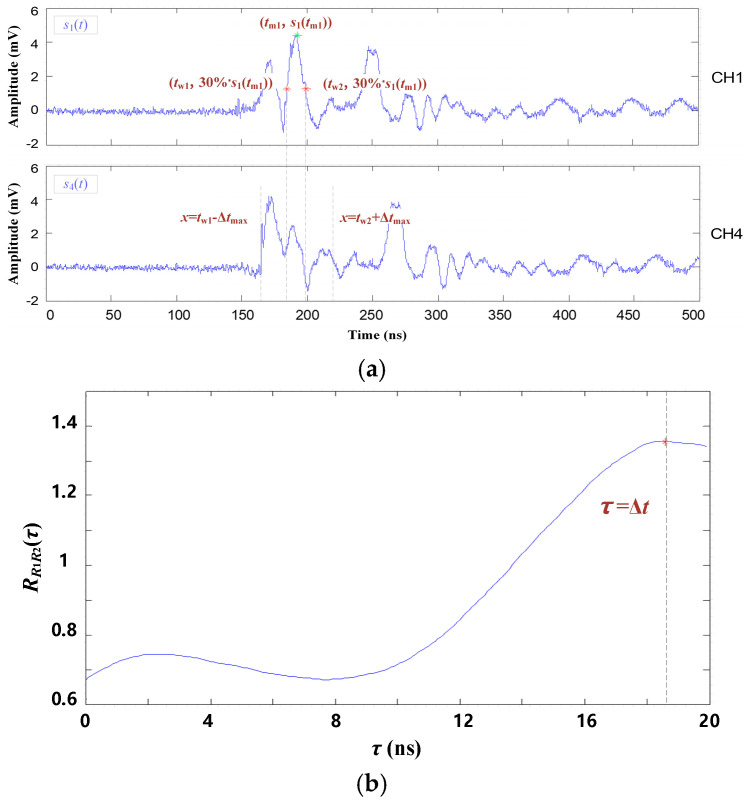
The secondary correlation of first characteristic signal segment. (**a**) The first characteristic signal segment; (**b**) Secondary correlation function and TDOA.

**Figure 10 entropy-25-00572-f010:**
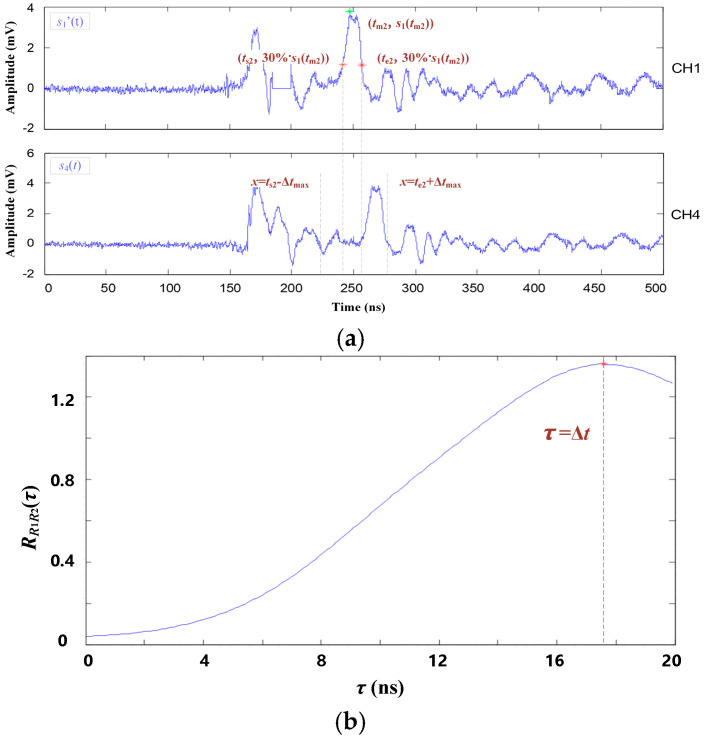
The secondary correlation of the second characteristic signal segment. (**a**) The secondary characteristic signal segment; (**b**) Secondary correlation function.

**Figure 11 entropy-25-00572-f011:**
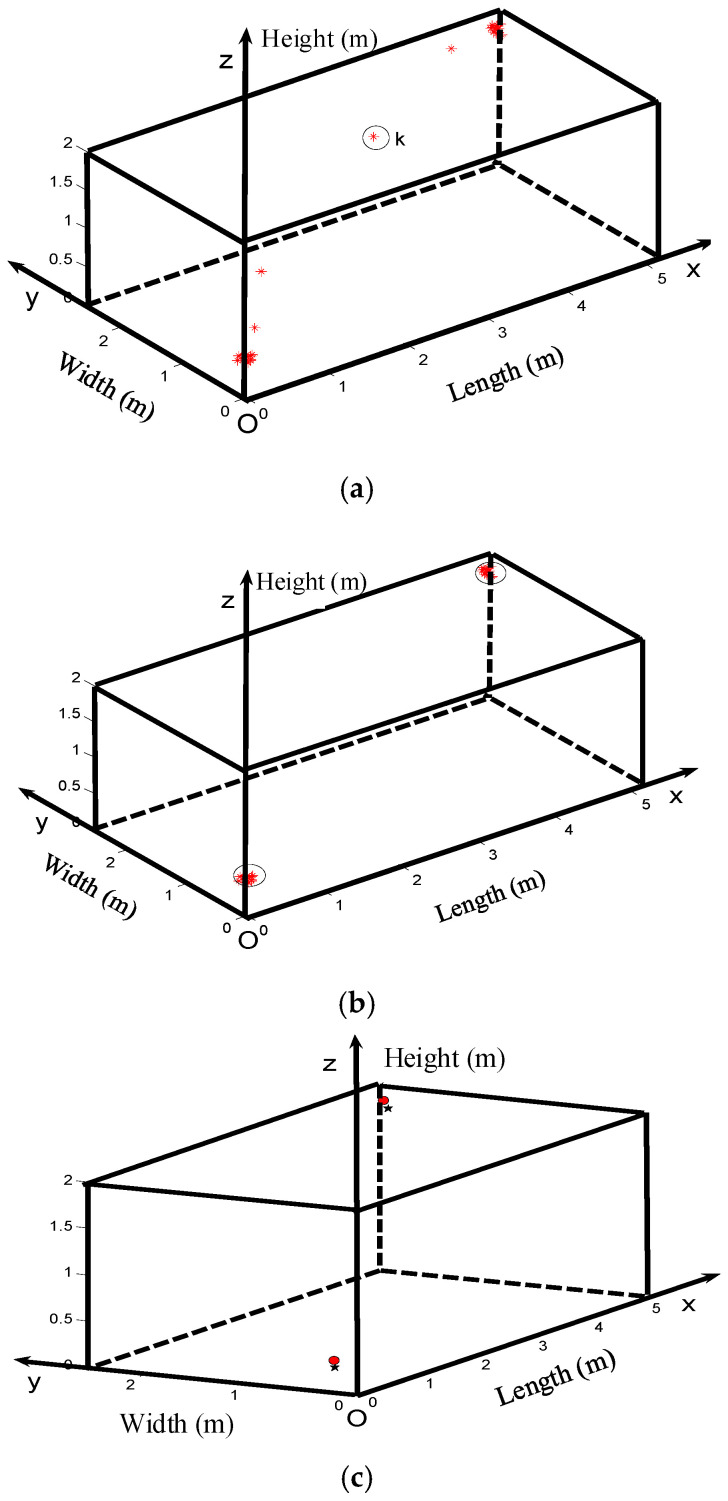
PD source calculated by dynamic weighting. (**a**) Point density estimation; (**b**) Linear classification and dynamic weighting; (**c**) Location results and actual PD source.

**Figure 12 entropy-25-00572-f012:**
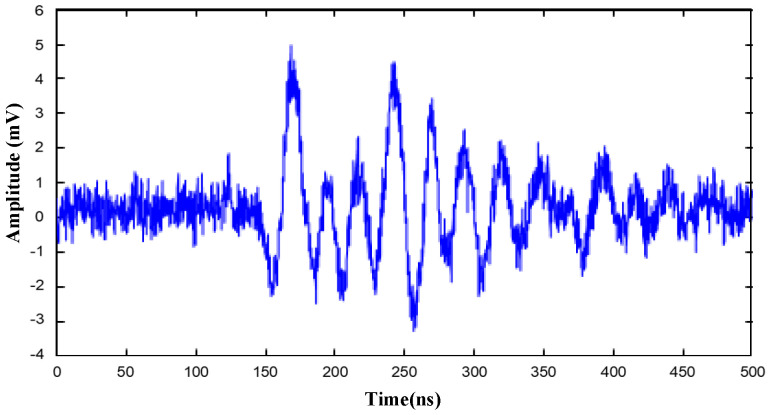
UHF partial discharge signals in the second set of experiments.

**Table 1 entropy-25-00572-t001:** TDOA extraction methods comparison.

	Error (ns)	First Peak Method	Secondary Correlation Method	Optimized Energy Accumulation Method
Theoretical TDOA (ns)				
1.6		0.16	0.47	0.19
7.3		0.68	0.48	0.45
11.0		1.35	0.30	0.17
16.8		1.26	0.52	0.43

**Table 2 entropy-25-00572-t002:** TDOAs obtained by the optimized energy accumulation method.

	TDOA (ns)	*P*_1_ (0.2 m, 0.3 m, 0.2 m)			*P*_2_ (5 m, 2.4 m, 1.8 m)		
Group		Δ*t*_12_	Δ*t*_13_	Δ*t*_14_	Δ*t*_12_	Δ*t*_13_	Δ*t*_14_
Theoretical value		15.3	0.7	18.0	−16.6	−2.4	−17.5
1		15.3	0.6	18.1	−18.0	−3.8	−19.5
2		15.3	0.7	18.1	−17.8	−3.2	−18.7
3		-	-	-	−15.7	−2.1	−16.6
4		-	-	-	−17.6	−3.4	−18.6
5		15.7	0.7	18.5	−16.9	−2.7	−17.8
6		15.3	0.5	18.1	-	-	-
7		14.8	0.6	17.0	−16.8	−2.0	−18.7
8		-	-	-	−17.4	−3.5	−18.8
9		15.4	0.7	18.2	−17.1	−2.9	−18
10		14.8	0.4	16.7	−18.0	−3.2	−19.2
11		14.6	0.5	16.5	−18.0	−3.7	−17.2
12		-	-	-	−17.2	−2.2	−18.8
Average absolute error		0.28	0.11	0.60	0.88	0.74	1.07

**Table 3 entropy-25-00572-t003:** TDOAs obtained by the secondary correlation method.

	TDOA (ns)	*P*_1_ (0.2 m, 0.3 m, 0.2 m)			*P*_2_ (5 m, 2.4 m, 1.8 m)		
Group		Δ*t*_12_	Δ*t*_13_	Δ*t*_14_	Δ*t*_12_	Δ*t*_13_	Δ*t*_14_
Theoretical value		15.3	0.7	18.0	−16.6	−2.4	−17.5
1		15.0	0.6	19.5	−16.2	−2.4	−17.0
2		15.3	0.7	18.0	−15.7	−1.5	−16.6
3		15.7	0.5	19.0	−17.8	−3.2	−18.8
4		15.0	0.1	17.2	−17.0	−2.1	−18.0
5		14.3	0.5	17.1	−18.0	−2.4	−18.9
6		14.8	0.3	17.0	−15.1	−1.7	−18.3
7		15.4	0.7	18.5	−16.0	−2.1	−18.9
8		14.7	0.5	16.9	−17.5	−3.0	−18.9
9		14.6	0.5	16.5	−17.0	−2.9	−17.0
10		15.0	0.6	18.5	−18.0	−3.6	−18.0
Average absolute error		0.42	0.21	0.88	0.91	0.53	0.99

**Table 4 entropy-25-00572-t004:** Location results and error.

Method	PD Source (cm)	Location Result (cm)	Error (cm)
Optimized energy accumulation method	*P*_1_ (20, 30, 20)	(25.5, 36.7, 24.9)	10.0
	*P*_2_ (500, 240, 180)	(494.7, 233.5, 178.7)	8.5
Secondary correlation method	*P*_1_ (20, 30, 20)	(29.8, 28.1, 26.9)	12.1
	*P*_2_ (500, 240, 180)	(503.7, 235.5, 185.2)	8.4
	*P*_3_ (220, 135, 55)	(233.7, 141.0, 52.4)	15.2
	*P*_4_ (285, 155, 50)	(272.1, 148.0, 52.0)	14.8

## Data Availability

No new data were created or analyzed in this study. Data sharing is not applicable to this article.

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
