# Peer review of "Multi-Source Partial Discharge Fault Location with Comprehensive Arrival Time Difference Extraction Method and Multi-Data Dynamic Weighting Algorithm"

_entropy, 2023, doi:10.3390/e25040572_

Round 1

Reviewer 1 Report

entropy-2235514

Multi-Source PD Fault Location with Comprehensive TDOA Extraction Method and Multi-data Dynamic Weighting Algorithm

To localize partial discharge fault, authors proposed a hybrid method of time difference of arrival (TDOA) extraction and multi-data dynamic weighting algorithm. This method achieves higher accuracy than those of other methods. Generally, the novelty of the manuscript is good to support it for publication. However, I propose several questions for the author to explain, so as to effectively improve the quality of this manuscript. Specific comments are as follows:

--In the title, Abbreviations should be avoided for they are not well-known for readers. 

--Why and how to use wavelet transform? Please give more explanations about this point.

--Eq.(4) might not correctness, please check it. Also, please carefully check the formulas you given in the manuscript.

-- Photos of Experiment platform as well as the measurement systems should be given.

--English should be improved by correcting some grammar errors as well as informal expressions, such as ‘dynamic weighting algorithm’ and ‘dynamic weighting operations’ should be the unique one, etc.

--About the application strategy, I think a more concise description should be given points by points.

Author Response

First, we would like to thank the editor for his/her consideration and recommendation. Also we would like to thanks for the reviewer. The aspects pointed out by the reviewer are appropriate and the answers and changes to all of them have been included in the revised version of the paper. Based on the comments and suggestions provided by the reviewers, we deeply revised our paper and we are resubmitting a new version for review with a list of the updates that we made from the previous manuscript. All changes have been marked as blue.

Reviewer 2 Report

To extract time delay from waveform with heavy noise is an interesting and challenging topic. The current research uses such technique to diagnose discharging problems of equipment in electrical grid, giving reasonable testing result. To make it more attractive to the scientific community, some questions should be considered:

1. The signal shown in the provided figures does not appear to have a very low SNR. Can it be shown that the proposed method has good noise reduction capability?

2. Is there any difference between the PD source used in the simulation experiment and the PD source inside the real electrical equipment?

3. Does the presence of internal structures in electrical equipment, such as iron cores, render the above methods completely unusable?

4. Strengthen the description of theoretical formulas and parameters to make the technical content more accessible. Such as page 5 line 173, How does E[Rss(t-â–³t)Rsn1(t+t2)]“ come about?

Author Response

(The authors gave the same response as above.)

Round 2

Reviewer 1 Report

No further comments.

Reviewer 2 Report

The authors have revised the manuscript based on the previous comments, it can be accept now.